# Mind the gender gap: COVID-19 lockdown effects on gender differences in preprint submissions

**Iñaki Ucar**[1☉]*, **Margarita Torre**[2☉], **Antonio Elías**[3☉]

**1** UC3M-Santander Big Data Institute, Universidad Carlos III de Madrid, Getafe, Spain, **2** Department of Social Sciences, Universidad Carlos III de Madrid, Getafe, Spain, **3** OASYS group, Universidad de Málaga, Málaga, Spain

☉ All these authors are contributed equally to this work.
* inaki.ucar@uc3m.es

**Data Availability Statement:** The complete dataset is archived on Zenodo (https://doi.org/10.5281/zenodo.5142676). R code used to analyse the data is archived at https://github.com/CONCIERGE-CM-UC3M/COVID19-gender-gap.

## Abstract

The gender gap is a well-known problem in academia and, despite its gradual narrowing, recent estimations indicate that it will persist for decades. Short-term descriptive studies suggest that this gap may have actually worsened during the months of confinement following the start of the COVID-19 pandemic in 2020. In this work, we evaluate the impact of the COVID-19 lockdown on female and male academics' research productivity using preprint drop-off data. We examine a total of 307,902 unique research articles deposited in 5 major preprint repositories during the period between January and May each year from 2017 to 2020. We find that the proportion of female authors in online repositories steadily increased over time; however, the trend reversed during the confinement and gender parity worsened in two respects. First, the proportion of male authors in preprints increased significantly during lockdown. Second, the proportion of male authors in COVID-19-related articles was significantly higher than that of women. Overall, our results imply that the gender gap in academia suffered an approximately 1-year setback during the strict lockdown months of 2020, and COVID-related research areas suffered an additional 1.5-year setback.

## Introduction

The under-representation of women in scientific publications is well established in the literature. Despite their growing presence in all research areas, women continue to publish less than men [1–3], even in fields where they are not a minority [4]. In addition, women are much less likely to be included as first or last authors in an article [3, 5]. According to recent estimates by Holman and his colleagues [6], a significant gender gap will persist for decades, especially in the areas of computer science, physics, and mathematics. As a result, women are less likely than men to be granted tenure or promoted [1, 7, 8].

How has the gender gap in scientific production changed during the global lockdown? And how might these changes affect the career paths of men and women in the short and medium term? Here, we build on recent research to argue that the move towards gender parity could

**Funding:** This work has been supported by the Madrid Government (Comunidad de Madrid) under the Multiannual Agreement with UC3M in the line of "Fostering Young Doctors Research" (CONCIERGE-CM-UC3M), and in the context of the V PRICIT (Regional Programme of Research and Technological Innovation. M.T. and I.U. also acknowledge support from the Spanish Ministry of Science and Innovation through the research grant RTI2018-098182-A-I00.

**Competing interests:** The authors have declared that no competing interests exist.

slow down as a result of COVID-19. On the one hand, surveys examining time allocation and research production during the pandemic suggest that research productivity has decreased more for women, particularly those with young children [9–11]. On the other hand, research funds are being redirected to support coronavirus-related studies, even at the cost of other less cutting-edge topics. Those with the ability to initiate COVID-related research projects will be more likely to benefit from these new lines of funding. If the majority of scholars conducting such novel research are men, then they will have gained a further advantage in the coming years. All in all, lower levels of scientific production during confinement could be detrimental to women—particularly those in early career stages [12]—who may see diminished promotion possibilities or even risk losing their jobs.

Assessing the magnitude and scope of the new gap in scientific production is, therefore, crucial to designing and implementing effective actions that will prevent a backslide in academic gender equity. With this aim, we model the evolution of the gender gap in preprint submissions from January 2017 to May 2020 to measure the impact of COVID-19 on male and female scientific productivity. Specifically, we examine a total of 307,902 unique research articles deposited in 5 major repositories: arXiv, medRxiv, bioRxiv, PsyArXiv, and SocArXiv. A preprint is a full draft article that is shared publicly before it has been peer reviewed. Preprints offer strong benefits, such as the possibility of receiving feedback and increased visibility, which often results in a higher number of citations. In addition, during the COVID-19 pandemic, they have enabled researchers to share data and essential findings at unprecedented speeds. We aim to quantify the effect of confinement on the likelihood of men and women to publish a preprint and, more specifically, to publish COVID-related research.

Our research contributes to prior studies in several ways. First, unlike previous research focusing on a very limited period [13–15], we examine preprint submission trends from 2017 to 2020. By expanding the observation window, we are able to discern how much of the observed change in 2020 corresponds to the effect of lockdown and social distancing and how much is due to gender differences in the circulation of scientific knowledge over time [16–18]. Second, unlike previous studies, we provide a systematic analysis of academic fields. Some scholars have examined very broad areas of knowledge, such as mathematics, physics, or economics [13–15], while others have focused on journals on a very specific topic, such as medicine [19, 20] or biological sciences [21]. Our study seeks to fill this gap by covering a total of 10 academic fields and 250 sub-fields. Thus, we provide a comprehensive view of the pandemic's impact on scientific productivity. Also, as we will see, this level of granularity is crucial to avoid incurring Simpson's paradox, where group trends disappear or reverse when data is aggregated. Third, our model discriminates between COVID-19-related research and general research. While the coronavirus has brought many challenges to academic research, it has also created opportunities for research moving forward. Therefore, it is key to look at who is taking advantage of these new opportunities.

Finally, we pay detailed attention to authorship order. Given the growing tendency across scientific disciplines to write multi-author papers, the sequence of names is becoming a major topic in recruitment processes, promotion, and tenure [3, 22, 23]. Thus, we not only examine whether gendered patterns of authorship vary after lockdown among all authors but also among solo authors, first authors, and last authors. It is common practice across many areas that the first author contributes most to the work and receives the most credit. Therefore, given the asymmetric share of domestic responsibilities women have assumed during confinement [24–26], we expect a decrease in females listed as first authors and as solo authors, the two positions requiring the most intensive research work. This decline might be particularly noticeable in areas of knowledge where women have recently joined, since young female academics are more likely to have kids and experience the caregiver burden. As for the other

authorship positions, the pre-lockdown expectations are not so clear. In some disciplines, the last author position is reserved for the senior author (first-last-author-emphasis norm, or FLAE), while in other fields the author sequence reflects their relative contributions to the manuscript (sequence-determines-credit approach, or SDC) [23]. Consequently, variations in gender composition after lockdown cannot be easily anticipated.

## Data and methods

The complete dataset is publicly available on Zenodo [27], where the software repository containing the replication scripts is linked as supplementary materials.

### Data collection and integration

Using their APIs, we collected public data from the top 5 preprint servers in terms of submissions with availability since January 2017 (namely, arXiv, bioRxiv, medRxiv, PsyArXiv, and SocArXiv). We used R [28], and, specifically, the following packages: aRxiv [29], medrxivr [30] and osfr [31]. We downloaded all submission records from January to May for the years 2017, 2018, 2019, and 2020. The arXiv API returns the date of the last update for preprints with more than one submission. For the other repositories, the date is defined as the first posting date. Specifically, we collected information on full author names, order of appearance, publishing date, categories, subcategories, article title, abstract, and keywords. We restricted the analysis to the March-May period, when the lockdown was considerably uniform among countries. When contagions dropped during June and July, internal and external border restrictions were relaxed. However, the restrictions were lifted unevenly and the confinement situation became more heterogeneous across countries and continents, potentially skewing the data. In addition, focusing on the short-term period allows us to capture gender differences when researchers are put under time pressure and must carry out their work in challenging circumstances.

### Preprint categorization

Different preprint servers require different approaches to preprint categorization. On the one hand, the arXiv repository consists of 8 categories (such as Computer Science, Mathematics, or Physics) and a reduced set of subcategories for large subject areas within those main categories. bioRxiv and medRxiv follow a similar approach for the Biology and Health Sciences categories, respectively. On the other hand, PsyArXiv and SocArXiv (Psychology and Social Sciences) allow the authors to freely tag submissions with a (potentially unlimited) number of areas, sub-areas, and even more specific fields of study. As a result, compared to the rest of the repositories, PsyArXiv and SocArXiv contain a large number of subcategories for a relatively small number of preprints, as shown in S1 Table.

Thus, to better balance our categorization, we pre-processed PsyArXiv and SocArXiv data to consider just the preprint's *main subcategory*. To identify this main subcategory, we first sorted all the unique tags in descending order based on the number of papers using them. The main subcategory for each preprint was then defined as the first appearance in the previous list, and the rest of the tags were removed. We also discarded those subcategories with fewer than 100 preprints, and manually recoded some SocArXiv subcategories that were still too specific (see S2 Table). Finally, arXiv categories "Quantitative Biology" and "Quantitative Finance" were recoded and merged into Biology and Economics, respectively, and the Psychology subcategory contained in SocArXiv was merged into the Psychology category.

The summary after this pre-processing is shown in Table 1. While public repositories have gained popularity in all fields of knowledge, the table reveals that significant differences among

**Table 1. Number of subcategories and preprints per category after pre-processing.** In the arXiv repository, preprints are sometimes cross-tagged in several categories. As a result, the number of unique preprints is 307,902.

| Category | # subcategories | # preprints |
|---|---|---|
| Biology | 34 | 44454 |
| Computer Science | 40 | 79653 |
| Economics | 12 | 2869 |
| Elec. Eng. Systems Science | 4 | 10686 |
| Health Sciences | 52 | 6100 |
| Mathematics | 32 | 78187 |
| Physics | 51 | 116983 |
| Psychology | 14 | 4120 |
| Social Sciences | 5 | 1938 |
| Statistics | 6 | 22040 |
| Total: | 250 | 367030 |

disciplines persist. Physics, Computer Science, and Mathematics are the three areas with the highest number of submissions. At the other extreme are Economics, Social Science, and Psychology. These figures are consistent with previous research showing that journals from STEM disciplines have clearer policies regarding preprinting than journals from the Social Sciences and Humanities, which could affect authors' decisions to submit their work to open-access repositories [32].

Next, a data quality assessment was conducted to detect and remove possible inconsistencies. First, we processed full names to remove stop words, places, and institutions, so that, for example, an author's institution did not register as an additional author. Second, we removed articles with inconsistent dates. Finally, because in some repositories supplementary materials appeared as an additional posting, we removed them, as well as preprints that were marked as withdrawn from the repository.

## Inferring gender from authors' given names

We used the genderize.io database to assign gender to authors based on their first names. This is one of the most effective gender prediction tools [33] and has been widely used in the literature ([6], among others). One of the most important advantages of using name-to-gender inference services is that, compared to standard approaches for name-to-gender inference based on administrative data (census data, administration records, or country-specific birth lists), they allow for a robust prediction for names from countries all over the world (see [34] for an evaluation of different web services). Our choice, *Genderize*, is a database of name–gender associations assembled from all over the web (> 114M given names for approximately 80 countries as of January 2021), and thus is a good option for analyses on data outside of a national context. Gender data was collected via *Genderize*'s API using the `genderizeR` R package [35] for a total of 1,235,037 unique authors.

In addition to predicting gender for a given name, *Genderize* returns additional information to quantify the precision of such predictions, namely, *count* and *probability*. The count shows how many instances in the database associate a given first name with the predicted gender, and the probability corresponds to the proportion, or frequency, of such associations. Unfortunately, gender cannot be predicted when the authors' given names were written as initials or were absent from the *Genderize* database, and these instances were reported as missing cases. The *Genderize* API reported a total of 28% of missing cases. Additionally, following

**Table 2. Preprints considered per year and model.** Proportion (*p*) and number (*N*) of preprints included in the analyses for each year and model after filtering out missing cases.

| Model | 2017 | | 2018 | | 2019 | | 2020 | |
|---|---|---|---|---|---|---|---|---|
| | *p* | *N* | *p* | *N* | *p* | *N* | *p* | *N* |
| all authors | 0.76 | 39843 | 0.78 | 49623 | 0.80 | 60939 | 0.79 | 91030 |
| first author | 0.66 | 13941 | 0.65 | 19002 | 0.65 | 24857 | 0.64 | 37993 |
| last author | 0.74 | 15379 | 0.73 | 21252 | 0.73 | 27765 | 0.73 | 43014 |
| single author | 0.78 | 6605 | 0.77 | 7010 | 0.77 | 7328 | 0.76 | 9965 |

Holman and his colleagues [6], we only consider gender identification with a probability higher or equal to 0.95 and a frequency of at least 10 appearances in the *Genderize* database. This simple procedure preserved 80% of cases. After filtering out missing cases, the proportion of preprints included in the analyses were equivalent for all the years (see Table 2), indicating that this procedure did not introduce any under- or over-representation bias.

## Measuring the effect of lockdown

We are interested in estimating the gender gap in preprint submissions and measuring how much of this gap can be attributed to the global lockdown. Previous research examining gender differences in publications has mostly used generalized linear models (GLMs) to estimate the gender proportion and its rate of change ([6, 36], among others). However, this approach takes the averages of the individual-level variables, discarding valuable within-group information that may reveal opposing trends. A potential alternative would be to disaggregate and introduce categories and subcategories as fixed effects in the GLM design, but this would violate the assumption of independence of the observations, thus biasing the results. To account for this drawback, we employ a hierarchical (or multilevel) GLM, which explicitly models the nested nature of this data.

More concretely, we define a fractional hierarchical GLM model that captures the proportion of males as a function of time and measures the effect of the lockdown and type of research (directly related to COVID-19 versus not directly related), with a random intercept per category and subcategory. We consider the quasi-binomial family to describe the error distribution (to account for overdispersion) with the logit link function:

$$\text{logit}\,(p_{\text{male},i}) = \beta_0 + \alpha_{0,k[j[i]]} + \beta_1 \cdot \text{year} + \beta_2 \cdot \text{lockdown} + \beta_3 \cdot \text{COVIDpaper} + \epsilon_i$$

$$\alpha_{0,k} = \alpha_{1,j[i]} \cdot \text{category} + \eta_k \qquad (1)$$

$$\alpha_{1,j} = \alpha_{2,i} \cdot \text{subcategory} + \gamma_j$$

where *i*, *j*, and *k* index observations, subcategories, and categories, respectively; the terms $\epsilon_i$, $\gamma_j$, and $\eta_k$ are normal errors at the individual and cluster levels; the response variable $p_{\text{male}}$ is the proportion of males; 'year' is a continuous variable that takes a value of 0 at the start of our time window (i.e., 2017 → 0, 2018 → 1 and so on); 'lockdown' is a binary factor that is equal to 1 during the lockdown period, from March to May 2020; 'COVID paper' is a binary factor that is equal to 1 for preprints directly related to COVID-19, defined as those preprints containing "coronavirus", "sars-cov-2," or "covid-19" in their title (restricted to 2020 and with case-insensitive matching); and where we consider a random intercept that varies across categories and subcategories inside categories.

Within this framework, we consider four distinct models. Namely, we estimate the monthly proportion of males for (1) *all authors*, (2) *first authors*, (3) *last authors*, and (4) *single authors*. In all cases, $p_{\text{male}}$ is computed as the total number of males over the total number of males and

females identified per month, excluding missing values. In the case of *all authors*, preprints with missing gender rates greater than 25% are not considered, and subcategories with fewer than 30 authors per month are dropped too. In the case of *first* and *last authors*, preprints with an alphabetically-ordered list of authors are discarded, as alphabetical sequence is frequently used to acknowledge similar contributions or to avoid disharmony among collaborating groups [23]. In the case of *single authors*, only preprints with one author are considered.

## Results and discussion

### Gender trends in preprints submission over time

Global numbers in Fig 1 show that the trend of preprint submissions has accelerated notably over the previous three years, and especially during lockdown. This effect is particularly pronounced in fields where COVID-related production is more likely—such as biology, health sciences (vaccines, epidemiology, etc.), and mathematics (epidemiological models)—but is also clearly evident in computer science, economics, engineering, physics, and psychology. Also, the time trend in the social sciences and economics is less constant than in the other areas. This is largely due to the lower number of submissions registered in these fields. As we have seen in Table 1, both social sciences and economics rank at the bottom in terms of the number of preprints received, with $N = 1938$ and $N = 2869$, respectively, far behind other areas such as physics ($N = 116983$), computer science ($N = 79653$), and mathematics ($N = 78187$). In spite of these variations, results suggest that research in all areas has been very prolific during 2020, particularly during the lockdown months.

Next, Fig 2 displays the male proportion by category. In general terms, we do observe a slowly declining global trend. This is mainly driven by categories that are already more feminized than the average, such as biology, health sciences, and psychology. As for the rest of the categories, the gender gap has remained rather stable during the period considered, in

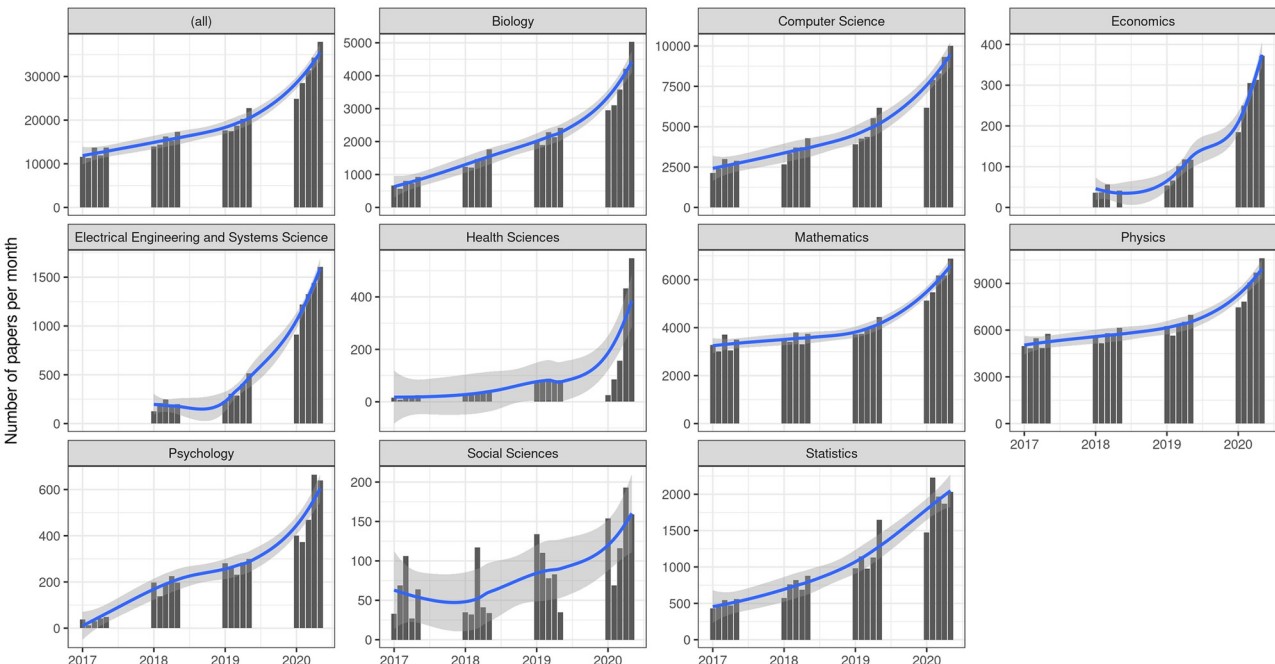

**Fig 1. Number of submissions per month.** The first facet (*all*) shows the global number of preprint submissions per month in all repositories during the period considered. Subsequent facets break down these numbers per category with varying scales for the vertical axis.

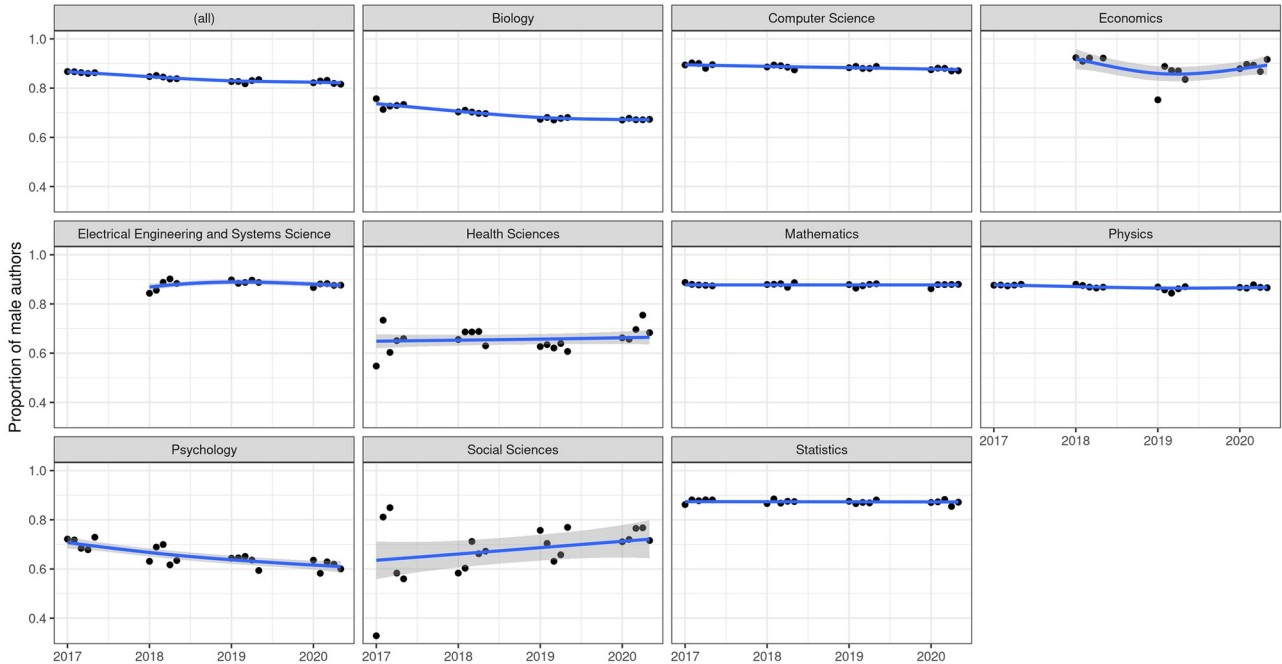

**Fig 2. Proportion of male authors per month.** The first facet (*all*) shows the global proportion of males per month submitting to all repositories during the period considered. Subsequent facets break down these numbers per category.

consonance with previous findings [6]. The social sciences are the only exception to the general trend, showing a slight upswing in the proportion of males. As discussed above, this oddity might be related to a less frequent and irregular use of online servers in this particular field of study.

Consistent with previous studies [14, 15], Fig 2 reveals a slowing down of the feminization process during the pandemic. This slowdown can be separately observed in biology and psychology, and the gender gap has even started to grow in health sciences and economics. However, it is difficult to conclude with certainty whether such an effect exists and, even if it does, whether it can be ultimately attributable to the lockdown period, as other authors have suggested from similar descriptive analyses. To better account for these trends, we next explore this issue in a hierarchical modelling framework.

## Explaining the gender gap

We estimate the monthly proportion of males for (1) all authors, (2) first authors, (3) last authors, and (4) single authors, as described in the Methods section. Our model separates the temporal trend from the effect of the lockdown, controlling for COVID-related work, and uses random intercepts to take into account the hierarchical nature of the data, which is nested in categories and subcategories. Table 3 shows the coefficient estimates and main summary statistics for all the models considered.

The use of a GLMM model is justified by comparing model (1) with model (0), which is a GLM for the same data but does not account for this hierarchical structure. There are three reasons that lead us to opt for the GLMM model. First, the GLMM achieves a much better fit and predictive power (83% of variance explained). Second, the model intercept correctly captures the overall average proportion of males at the beginning of our time window ($\sim 83\%$ of males as of January 2017, as can be seen in Fig 2). Finally, once we include categories and

**Table 3. Regression analysis results.** Table of coefficients and summary statistics for the models considered.

| | *Dependent variable*: | | | | |
|---|---|---|---|---|---|
| | **Proportion of males** | | | | |
| | **all authors** | **all authors** | **first author** | **last author** | **single author** |
| | *fractional GLM* | *fractional GLMM* | | | |
| | **(0)** | **(1)** | **(2)** | **(3)** | **(4)** |
| year | −0.104*** | −0.049*** | −0.044*** | −0.059*** | −0.029 |
| | (0.003) | (0.003) | (0.009) | (0.009) | (0.018) |
| lockdown | 0.072*** | 0.031*** | 0.035* | 0.008 | 0.136*** |
| | (0.007) | (0.007) | (0.020) | (0.021) | (0.052) |
| COVID paper | −0.614*** | 0.076*** | 0.399*** | 0.129** | 0.715** |
| | (0.019) | (0.022) | (0.064) | (0.065) | (0.298) |
| (Intercept) | 1.826*** | 1.595*** | 1.520*** | 1.819*** | 2.480*** |
| | (0.006) | (0.199) | (0.195) | (0.173) | (0.169) |
| Observations | 3,368 | 3,368 | 3,996 | 4,027 | 3,517 |
| N (subcategory) | | 192 | 201 | 201 | 200 |
| N (category) | | 10 | 10 | 10 | 10 |
| sd(subcategory) | | 0.29 | 0.32 | 0.30 | 0.45 |
| sd(category) | | 0.62 | 0.60 | 0.53 | 0.47 |
| Log Likelihood | −39,496.310 | −11,327.900 | −7,182.127 | −6,955.634 | −3,788.911 |
| Akaike Inf. Crit. | 79,000.620 | 22,667.800 | 14,376.250 | 13,923.270 | 7,589.823 |
| Bayesian Inf. Crit. | | 22,704.540 | 14,414.010 | 13,961.070 | 7,626.815 |
| Pseudo-$R^2$ | 0.04 | 0.83 | 0.24 | 0.18 | 0.11 |

*Note*:

*p<0.1;

**p<0.05;

***p<0.01

subcategories as random intercepts, we observe that the sign of the 'COVIDpaper' coefficient changes. This result reveals that disaggregation is necessary to avoid incurring a Simpson's paradox. Moreover, these random intercepts also correctly capture well-known differences among categories and subcategories. For example, we observe that STEM categories have a wider gender gap than the average, while non-STEM ones are much more feminized (see S1 Fig). Similarly, subcategories such as "high-energy" and "quantum physics" are more masculinized than the average for all physics, while astrophysics-related research as well as "bio-medical physics" are more feminized (see S2 Fig).

To facilitate model comparison, Fig 3 displays the relative effect size with 95% confidence intervals (CI) for the three fixed effects in (1–4). Results can be summarized in three key points. First, findings confirm a slow but significative decreasing trend in the overall proportion of male authors (OR 0.95, 95% CI 0.95–0.96); this holds for first (OR 0.96, 95% CI 0.94–0.97) as well as last authors (OR 0.94, 95% CI 0.93–0.96). The reduced number of single authors, however, does not provide enough statistical power to measure such a small effect (should it exist), but the point estimate is consistent with the estimates for the rest of the models (OR 0.97, 95% CI 0.94–1.01).

Second, we find that for all authors, there is a measurable lockdown effect that is slightly smaller than the yearly effect but with the opposite sign (OR 1.03, 95% CI 1.02–1.05): the lockdown has partially reversed the yearly decrease in the proportion of male authors that would be expected in 2020 given the trend from previous years. A very similar effect for first authors

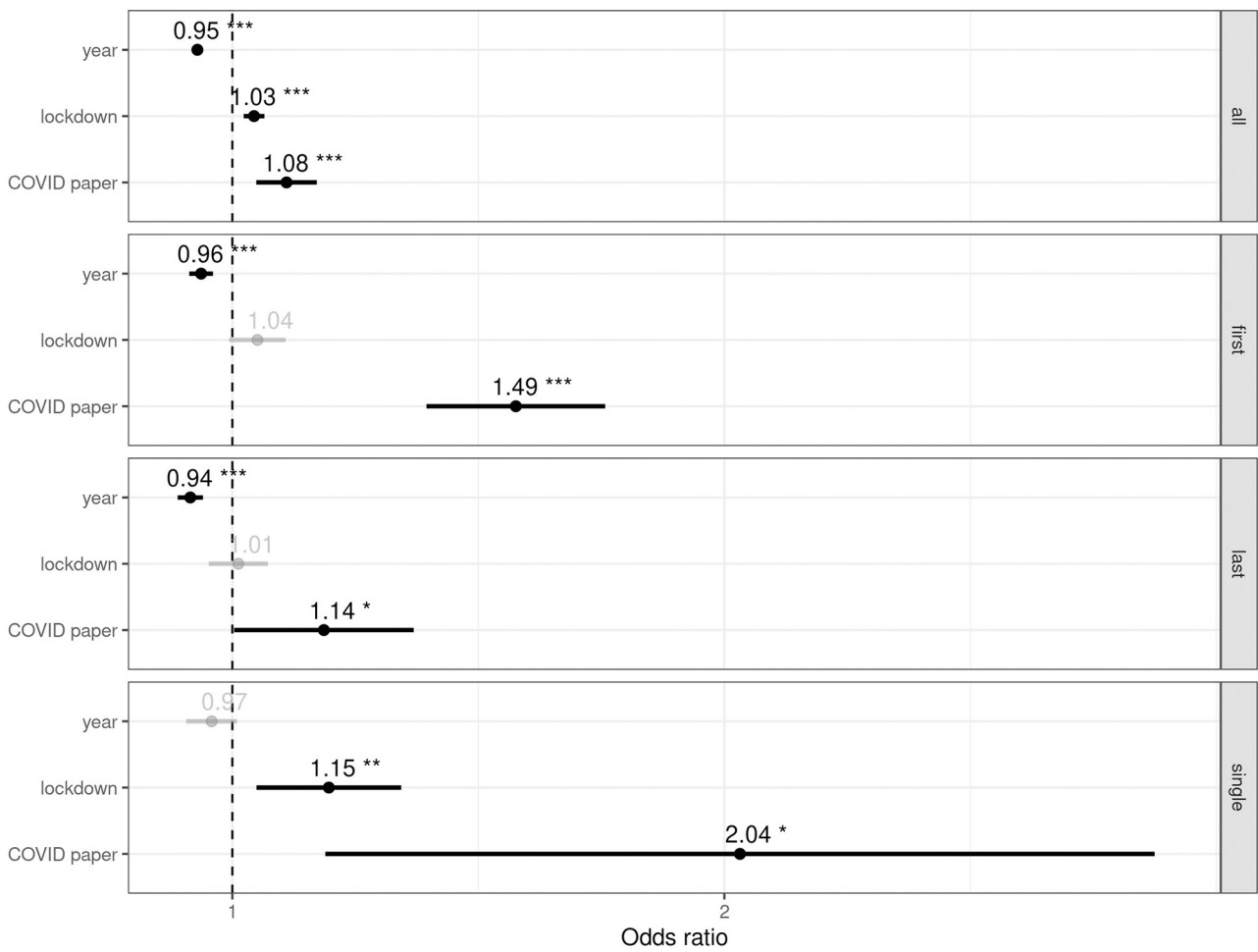

**Fig 3. Relative effect sizes.** Odds ratio with 95% CIs for the fixed effects in all the models considered, from top to bottom: (1) all authors, (2) first authors, (3) last authors, and (4) single authors.

could exist, although the variability is higher and thus there is not enough statistical power to reach a stronger conclusion (OR 1.04, 95% CI 1.00–1.08). Regarding single authors, a statistically significant and potentially much stronger effect can be observed (OR 1.15, 95% CI 1.03–1.27). This can possibly be explained by the greater effort and time required to produce single-authored papers, which could negatively affect women, especially those with children [24–26]. Finally, no effect is found for last authors (OR 1.01, 95% CI 0.97–1.05). This may be due to the fact that this co-authorship position has different meanings across disciplines. While in disciplines following the first-last-author-emphasis norm, last positions tend to be reserved for senior researchers with consolidated careers (e.g., senior female authors), the last position in the sequence-determines-credit approach is more likely to correspond to less-contributing authors (e.g., young women with child-rearing responsibilities). It seems reasonable to think that the latter may have had more trouble juggling child care and research than the former. These factors might result in a lower average value, and otherwise larger variability, for last author effects.

Third, we find an additional masculinization effect in COVID-related preprints that is equal or larger than the yearly feminization effect for all the models considered (*all authors*: OR 1.08, 95% CI 1.03–1.13; *first author*: OR 1.49, 95% CI 1.31–1.69; *last author*: OR 1.14, 95%

CI 1.00–1.29; *single author*: OR 2.04, 95% CI 1.14–3.67). In other words, overall, female authors' production has been penalized during lockdown compared to their male peers, but especially in those disciplines where increased productivity is directly linked to COVID-19 research. While the COVID-19 pandemic has resulted in unprecedented research opportunities worldwide, women have not benefited as much as men have. This is particularly noticeable among single authors and first authors, the two most time-consuming positions. As for the last authorship position, the relative effect size is lower but still significant.

## Conclusion

In this paper, we examine how the global lockdown has affected the gender productivity gap in academia. More specifically, we model the evolution of the gender gap in preprint submissions between 2017 an 2020. Our findings show that the progress towards gender parity that has been observed over the past few years partially reversed during the lockdown period. The pandemic confinements penalized women in two ways. Not only were they less likely to complete research during the pandemic, but they were also less likely to produce COVID-related research than men, despite the increasing research opportunities that COVID-19 provided in many fields. Overall, results indicate that the gender gap in academia suffered an approximately 1-year setback during the strict lockdown months of 2020, and COVID-related research areas (which incidentally have a better male-female balance) suffered an additional 1.5-year setback.

The results of our research are relevant from an empirical and substantive point of view. From the empirical perspective, our analysis indicates that dissagregated data at the sub-field level is key to untangling within-group trends that are otherwise concealed in global averages due to the profound gender gap difference that still exist across disciplines and sub-disciplines. In this regard, generalized linear mixed models are a natural choice to model the hierarchical structure of such data. From the substantive perspective, results show that COVID-19 lockdowns exacerbated gender inequalities such that their effects will be felt in the years to come.

Current differences in productivity levels might result in higher rates of gender inequality in the next few years. Negative effects in the short and medium term might be twofold. On the one hand, we expect a reallocation of research money at the expense of research areas funded prior to the pandemic, which can lead to an unequal distribution of resources. In addition, lower productivity levels will result in fewer citations, fewer research grants, and lower likelihood of promotion among women. While our work only examines the impact of COVID-19 lockdowns in academia, this prognosis might be valid for all women in high-skilled occupations where promotion tracks and human capital accumulation are crucial during early career years, to the point that early productivity declines might lead to job loss [37, 38]. Therefore, the implementation of gender equity actions is necessary in order to ensure that the COVID-19-related penalty does not translate into inequality in future recruitment and promotion processes.

## Supporting information

**S1 Fig. Random intercepts for categories for all authors (1).** The model captures the known trends for all the major categories. Economics, engineering, computer science, mathematics, physics, and, to a lesser extent, statistics are categories with a proportion of males over the global average. By contrast, social sciences, biology, health sciences, and, specially, psychology are more balanced than the average.
(PDF)

**S2 Fig. Random intercepts for subcategories for all authors (1).** Within each category, specific subcategories develop their own trends. For example, we observe that high-energy and quantum physics are more masculinized than the average for all physics, while astrophysics-related research and bio-medical physics are more feminized.
(PDF)

**S1 Table. Initial composition of the dataset.** Raw number of subcategories and preprints per category prior to any cleaning or pre-processing step.
(PDF)

**S2 Table. Manual adjustments for SocArXiv.** As detailed in Methods, the large number of tags added to PsyArXiv and SocArXiv documents required a separate methodology for pre-print categorization. As a final step of such methodology, SocArXiv also required manual recoding of the subcategories listed here.
(PDF)

## Author Contributions

**Conceptualization:** Iñaki Ucar, Margarita Torre, Antonio Elías.

**Data curation:** Iñaki Ucar, Antonio Elías.

**Formal analysis:** Iñaki Ucar, Margarita Torre, Antonio Elías.

**Funding acquisition:** Iñaki Ucar, Margarita Torre.

**Investigation:** Iñaki Ucar, Margarita Torre, Antonio Elías.

**Methodology:** Iñaki Ucar, Margarita Torre, Antonio Elías.

**Project administration:** Iñaki Ucar, Margarita Torre.

**Resources:** Iñaki Ucar, Margarita Torre, Antonio Elías.

**Software:** Iñaki Ucar, Antonio Elías.

**Supervision:** Iñaki Ucar, Margarita Torre.

**Validation:** Iñaki Ucar, Margarita Torre, Antonio Elías.

**Visualization:** Iñaki Ucar, Margarita Torre, Antonio Elías.

**Writing – original draft:** Iñaki Ucar, Margarita Torre, Antonio Elías.

**Writing – review & editing:** Iñaki Ucar, Margarita Torre, Antonio Elías.

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
