## [Decision Letter · Decision Letter 0]

22 Sep 2021

PONE-D-21-25133Mind the gender gap: COVID-19 lockdown effects on gender differences in preprint submissionsPLOS ONE

Dear Dr. Ucar,

Thank you for submitting your manuscript to PLOS ONE. After careful consideration, we feel that it has merit but does not fully meet PLOS ONE’s publication criteria as it currently stands. Therefore, we invite you to submit a revised version of the manuscript that addresses the points raised during the review process.

We look forward to receiving your revised manuscript.

Kind regards,

Alireza Abbasi

Academic Editor

PLOS ONE

Journal Requirements:

Additional Editor Comments (if provided):

Below are two reviews of your submission. Each reviewer raises important issues and concerns but each also sees potential. We invite you to resubmit a revissied version addressing the issues raised carefully. Good luck on the revision.

Reviewers' comments:

Reviewer's Responses to Questions

**Comments to the Author**

1. Is the manuscript technically sound, and do the data support the conclusions?

Reviewer #1: Yes

Reviewer #2: Partly

2. Has the statistical analysis been performed appropriately and rigorously? 

Reviewer #1: I Don't Know

Reviewer #2: No

3. Have the authors made all data underlying the findings in their manuscript fully available?

Reviewer #1: Yes

Reviewer #2: Yes

4. Is the manuscript presented in an intelligible fashion and written in standard English?

Reviewer #1: Yes

Reviewer #2: Yes

5. Review Comments to the Author

Reviewer #1: The paper analyses preprint submission to five major preprint services from 2017 to the early stages of lockdown in 2020 to find out if the pandemic had any impact on submission by female authors. The paper is timely and has interesting findings. The data set is large and seems to have been processed properly to deal with known issues in bibliographic data and gender identification. The authors have made the data available which is commendable.

I have a few comments and questions. My first question is about the dates chosen. I think May was the early stages of lockdown in 2020 and not sure why a later date wasn't chosen to allow enough time for the pandemic to have its impact on scholarly work.

Some of the gender bibliometric studies (see works by Mike Thelwall for instance) have used data such as US census or social security data for gender identification arguing other data are based on the social web and therefore, unreliable and usually lack transparency. does this issue apply to Genderize service, or it is a transparent and reliable source.

The paper hasn't covered the related literature properly. while I didn't expect to see coverage of all gender bibliometric studies as there are many of them, the relevant papers that had a similar topic (impact of the pandemic on scientific productivity) should've been consulted and used if useful in the paper. An easy way to find most of such studies is to look at the papers that have cited the key paper by Viglione.

https://scholar.google.com/scholar?cites=15300145317525011924&as_sdt=2005&sciodt=0,5&hl=en

Finally, I think at least the Zenodo (or a link) should be mentioned in reference 20 otherwise readers won't be able to locate the source. currently, it is just author names and the title.

Reviewer #2: The manuscripts looks the disparity of productivity across males and females researcher by analysing a sample of nearly 500,000 preprints deposited during the years 2017 and 2020.

Language of the paper is at certain places very speculative. For example, predicting that the gender gap is going to persist for decades. How do we know that it is going to persist for decades? What is our indication and how many decades are we talking about? Could it be for the next 8-10 decades? Could this vanish over the next few decades? This is currently very vague and rather speculative. There are other examples of speculative arguments in the introduction too, and I am not very comfortable with them, because I can’t neither confirm or challenge those statements.

Language of the paper has at certain places been made unnecessarily complicated. In the sample, there were perhaps more 2020 papers with male authors than female authors. It is unclear why this has to be presented in a probabilistic language: “men were slightly more likely than women to submit preprints during lockdown”. Also being “slightly more likely” is not consistent with the sentence before claiming that the gap has widened during 2020. It is important that authors look at these findings with neutrality and not predisposed with the idea that the gender disparity has to have worsened during pandemic.

“men were significantly more likely than women to submit COVID-related research” – How is this related to overall productivity of male and female researchers? Why Covid topic has been singled out as a measure of productivity?

The reviewer also notes that more than 3 million articles are overall published each year, whereas the sample used in this study uses 500,000 pre-print items distributed over four years. While there is no prohibitive issue with sampling from pre-prints in general, one should note that they are not necessarily representative of the overall research production. The issue especially becomes important when the difference found between male and female is slight and can change after considering a bigger picture (i.e., the full amount of research produced) or published papers. Also, we cannot ignore the fact that these are pre-prints after all, and it is not clear what portion of them translated to official publications. This is especially a concern for covid-related publications in 2020 where an avalanche of papers were deposited in mass during first months of pandemic and many of them never got accepted due to insufficient quality/rigor.

For what portion of articles, the gender for all authors (or at least first author) could be determined at the specified threshold of confidence (0.95) and criteria (full names and not initials)?

p. 4 “However, this approach takes the averages of the individual-level variables, discarding valuable within-group information that may reveal opposing trends.” Please clarify. What does this mean? Are we talking about the interaction between contributing variables but in a rather complicated language?

If the function in Eq 1 is a logit model, then why the distribution of error is normal? That is not consistent with a logit model specification.

Why is p the proportion of males? The data is coming from individual articles. So what one would expect is for example p=1 if the paper has a male author (or a male first author, depending on what analysis you’re running). This is not aggregate model, so what is the reason for using the term proportion?

The most confusing part of this equation is the alpha coefficients, which have not even been elaborated on. Why such a confusing specification? What is the reason for the additional error terms for alphas? Those are going to confound with the main error term and the reviewer cannot even see the justification or interpretation of those.

Also this analysis (the logit analysis) is less problematic when we consider first authorships because a paper either has a male first author or a female first author (which is binary and consistent with a logit model specification). Therefore, if the coefficient for “year” for example is negative then one can conclude that the proportion of male authors have been decreasing over time (Although I am still not convinced why this has to be inferred from such an indirect complicated way as opposed to just reporting the proportion from the sample). But when it comes to “all author” analysis, then I am not sure how the models treats this. Is this where the issue of “proportion” comes to play? You count the proportion of male authors on the paper? Are you only using the papers for which the gender of all authors could be determined with 0.95 confidence?

Again, considering that the issue of lockdown was not a thing until 2020, I do not see the point of doing such joint/multivariate analysis, whereas, the proportion of males/females that published in 2020 could have just been reported independently. I apologise if this comment might sound too direct, but could it be that authors had to force themselves to use a more sophisticated statistical approach because simple reporting of proportions/stats could have come across rudimentary? Again, I apologise and I am sure that authors understand that the nature of a review sometimes entails direct questions of this nature. There is just limited interaction between these variables especially for 2017-2019 and that makes the use of such joint model very questionable to the reviewer. So could you justify the use of this method (and not choosing to just simply report the split of authorship in 2020 pre-prints)? Does it have to be inferred indirectly from a multivariate probabilistic model?

In the absence of captions for Supplementary figures, the reviewer unfortunately has no idea how to interpret them.

6. PLOS authors have the option to publish the peer review history of their article (what does this mean?). If published, this will include your full peer review and any attached files.

Reviewer #1: **Yes: **Hamid R. Jamali

Reviewer #2: No

---

## [Author Response · Author response to Decision Letter 0]

28 Oct 2021

We would like to thank the reviewers for their time and valuable comments, which have been a great help in improving the paper. In the following, we first describe the main changes that we have introduced in the revised version of the manuscript, and then provide a response to each reviewer's comments, explaining how these comments have been addressed in the new version of the paper. For convenience, along with the revised version of the manuscript, we submit an alternate version with track changes enabled that highlights changes in blue and red, denoting whether we added or removed content.

# Summary of changes

- First, we have endeavored to emphasize both the relevance and the contribution of our paper. Also, the introduction of the revised manuscript now offers a more comprehensive view of previous research on this topic.

- Second, we explain in more detail the issues concerning our research design, particularly those concerning the observation window and the gender classification process. 

- Finally, we have realized that the empirical portion of the paper was not presented with the precision required and raised some questions about the methods we are using. Consequently, we have better explained our analytical decisions. Also, we have redrafted the article aiming to make the results clear and intelligible to these and other readers. 

By expanding the frame, presenting the main findings more simply and clearly, and more judiciously assessing the theoretical contribution of the results, we believe that we have clarified the paper’s contribution to our evolving understanding of men's and women's productivity during lockdown. We hope that the revised version meets the high standards and expectations of PLOS ONE. 

Below we respond to all the concerns in detail. 

# Responses to reviewer 1

> The paper analyses preprint submission to five major preprint services from 2017 to the early stages of lockdown in 2020 to find out if the pandemic had any impact on submission by female authors. The paper is timely and has interesting findings. The data set is large and seems to have been processed properly to deal with known issues in bibliographic data and gender identification. The authors have made the data available which is commendable.

We thank the reviewer for their careful reading of the manuscript and their encouraging comments.

> I have a few comments and questions. My first question is about the dates chosen. I think May was the early stages of lockdown in 2020 and not sure why a later date wasn't chosen to allow enough time for the pandemic to have its impact on scholarly work.

We definitely agree with Reviewer 1 that the observation window is a crucial aspect of the research design. Expanding the observation period would be feasible. However, after giving this issue a great deal of thought, we concluded that keeping the short-term analysis has important advantages for our aim. From a substantive perspective, it allows us to capture gender differences at work in the context of time pressures and challenging circumstances (one of the most important sources of gender occupational segregation).

From an empirical perspective, while the lockdown was considerably uniform among countries from March to May 2020, the restrictions were lifted unevenly from June onward. Contagions dropped during June and July to levels that permitted internal and external border restrictions to be relaxed. However, the de-escalation process was not uniform, and the situation among countries/continents became more heterogeneous. Since our data do not allow for controlling by country, it is crucial to reduce the observation window to the most homogeneous time.

In the revised version of the manuscript, we have further elaborated on these decisions in the introduction section.

> Some of the gender bibliometric studies (see works by Mike Thelwall for instance) have used data such as US census or social security data for gender identification arguing other data are based on the social web and therefore, unreliable and usually lack transparency. does this issue apply to Genderize service, or it is a transparent and reliable source.

Indeed, name-to-gender inference based upon querying large name repositories (e.g., censuses, administration records, or country-specific birth lists) is particularly suitable when working at the national level, or with a limited number of countries. For analyses outside of a national context, however, name-to gender inference services are a better choice, as they can usually handle a greater degree of name diversity. In the particular case of the "Genderize" service (see [1] below for an evaluation of different web services), the underlying data is collected from social networks across 79 countries and 89 languages. Although the service does not geo-localize names automatically, it does accept two optional parameters, location and language, for more qualified guesses.

To further ensure accuracy in gender estimation, we follow the same classification process as Holman et al. (2018) [2] in this journal. When predicting gender for a given name, "Genderize" service returns additional information to quantify the precision of such prediction, namely, "count" and "probability". The count shows how many instances in the database associate a given first name with the predicted gender, and the probability corresponds to the proportion of such associations. Following [2], we only consider the drafts with a probability higher or equal to $0.95$. Also, we excluded names that were not found more than $10$ times in the "Genderize" database. 

We have added these details to the methods section in the revised version of the manuscript. Thank you for the comment. 

> The paper hasn't covered the related literature properly. while I didn't expect to see coverage of all gender bibliometric studies as there are many of them, the relevant papers that had a similar topic (impact of the pandemic on scientific productivity) should've been consulted and used if useful in the paper. An easy way to find most of such studies is to look at the papers that have cited the key paper by Viglione.

https://scholar.google.com/scholar?cites=15300145317525011924&as_sdt=2005&sciodt=0,5&hl=en

In the previous draft, we cited Viglione's paper but did not explore articles that cited it. Following the reviewer's suggestion, we have endeavored to make a more comprehensive review of the literature on COVID-19 and scientific productivity. As a result, the following references have been added: [3-10] listed below.

> Finally, I think at least the Zenodo (or a link) should be mentioned in reference 20 otherwise readers won't be able to locate the source. currently, it is just author names and the title.

Thank you for pointing this out. A wrong BibTeX declaration on our side led to the URL not appearing. This is fixed in the revised version of the manuscript (see reference 27).

# Responses to reviewer 2

We appreciate the reviewer's engagement with this research and the many specific and constructive suggestions for improvement.

> The manuscripts looks the disparity of productivity across males and females researcher by analysing a sample of nearly 500,000 preprints deposited during the years 2017 and 2020.

Language of the paper is at certain places very speculative. For example, predicting that the gender gap is going to persist for decades. How do we know that it is going to persist for decades? What is our indication and how many decades are we talking about? Could it be for the next 8-10 decades? Could this vanish over the next few decades? This is currently very vague and rather speculative. There are other examples of speculative arguments in the introduction too, and I am not very comfortable with them, because I can’t neither confirm or challenge those statements.

After reading the reviewer's comment, we realize that the introductory section of the manuscript has led to some confusion about the goal and contribution of our research. Our paper does not aim to estimate how long the gender gap of publications will last. The statement mentioned by the reviewer corresponds to a paper written by Holman et al. (2018), published in PLOS Biology [2]: "The gender gap in science: how long until women are equally represented?". In that paper, the authors affirm that "Topics such as physics, computer science, mathematics, surgery, and chemistry had the fewest women authors, while health-related disciplines like nursing, midwifery, and palliative care had the most. Of the gender-biased disciplines, almost all are moving towards parity, though some are predicted to take decades or even centuries to reach it" (section "The changing nature of the gender gap", first paragraph).

Here, we build on Holman et al. (2018) and other recent research (Viglione 2020, Myers et al. 2020, among others) to argue that the move towards gender parity could slow down as a result of COVID-19. We model the evolution of the gender gap in preprint submissions to measure the impact of COVID-19 on male and female scientific productivity during strict confinement. Overall, our results show that the gender gap in academia suffered an approximately 1-year setback during the strict lockdown months of 2020, and COVID-related research areas suffered an additional 1.5-year setback.

In the new version of the manuscript, we have revised the introductory section to clarify what corresponds to our article and what corresponds to previous research. Thank you for the remark.

> Language of the paper has at certain places been made unnecessarily complicated. In the sample, there were perhaps more 2020 papers with male authors than female authors. It is unclear why this has to be presented in a probabilistic language: "men were slightly more likely than women to submit preprints during lockdown". Also being "slightly more likely" is not consistent with the sentence before claiming that the gap has widened during 2020. It is important that authors look at these findings with neutrality and not predisposed with the idea that the gender disparity has to have worsened during pandemic.

Thank you for the observation. Following the reviewer's suggestion, we have redrafted the whole paper using a more direct language, endeavoring to communicate our meaning in a clear and concise manner. We hope these revisions help the reader. 

We would also like to stress that neither the findings nor their interpretation reflect our personal views or preferences. 

> "men were significantly more likely than women to submit COVID-related research" – How is this related to overall productivity of male and female researchers? Why Covid topic has been singled out as a measure of productivity?

While the coronavirus has created many challenges to conducting academic research, it has also created new opportunities (sometimes at the cost of less cutting-edge topics). As argued in the introduction, those (men or women) with the ability to launch COVID-related research projects will be more likely to benefit from these new lines of funding. Therefore, it is crucial to assess who is taking advantage of these new opportunities and whether there is a gender disparity among such researchers. 

In our analysis, 'COVID paper' is defined as a binary factor that equals 1 for preprints directly related to COVID-19, defined as those preprints containing "coronavirus," "sars-cov-2," or "covid-19" in their title (restricted to 2020 and with case-insensitive matching). Please see the methods section for a more detailed description. 

Therefore, we don't use COVID topics as a measure of productivity but as a key control in the analysis. In fact, a major contribution of our research is that COVID-related research during lockdown was signifcantly more masculinized than other kinds of research. 

> The reviewer also notes that more than 3 million articles are overall published each year, whereas the sample used in this study uses 500,000 pre-print items distributed over four years. While there is no prohibitive issue with sampling from pre-prints in general, one should note that they are not necessarily representative of the overall research production. The issue especially becomes important when the difference found between male and female is slight and can change after considering a bigger picture (i.e., the full amount of research produced) or published papers. Also, we cannot ignore the fact that these are pre-prints after all, and it is not clear what portion of them translated to official publications. This is especially a concern for covid-related publications in 2020 where an avalanche of papers were deposited in mass during first months of pandemic and many of them never got accepted due to insufficient quality/rigor.

We fully agree with the reviewer's observation that preprints and publications are not equivalent, and we would never make this claim. We contend, however, that an analysis of preprints is itself of interest. On the one hand, preprints are a productivity proxy increasingly exploited in the literature (see, for example, Viglione [11] or Matthews [12], among others). Their relevance is increasing, both in substantive and numeric terms. As we can see in Figure 1 in the manuscript, there is a growing body of scientists uploading their research papers to public repositories. Not surprisingly, journals in many fields are developing clearer policies regarding preprinting [13]. On the other hand, preprints are useful for assessing men’s and women’s ability to carry out research under challenging circumstances and time pressures, regardless of whether this work later translates into publications.

We also agree with the reviewer that the period under study is a changing scenario. Consequently, our analyses examine both the pre- and post-pandemic periods. Indeed, looking at the data from both of these periods is a major contribution to previous research.

> For what portion of articles, the gender for all authors (or at least first author) could be determined at the specified threshold of confidence (0.95) and criteria (full names and not initials)?

Thank you for the question, which we have addressed by adding a new table (see Table 2 in the revised manuscript) to the expanded section "Inferring gender from authors' given names". This new table displays the portion and number of preprints that entered the modelling stage after our filtering process due to missing gender identification cases. As can be seen, the proportion of preprints per year is quite high and, more importantly, it is approximately constant over time, which means that the filtering process did not introduce any under- or over-representation bias.

> p. 4 "However, this approach takes the averages of the individual-level variables, discarding valuable within-group information that may reveal opposing trends." Please clarify. What does this mean? Are we talking about the interaction between contributing variables but in a rather complicated language?

Please note that this paragraph does not refer to the interaction of contributing variables but to the necessity of using random intercepts in our analysis. More concretely, we discuss the difference between conventional Generalized Linear Models and hierarchical modelling, also called "multilevel" models. In the typical example using students' grades, students cluster within schools; similarly, here authorships cluster within categories and subcategories. In a GLM setting, group averages (e.g., category averages) are considered. This conceals trends for each group and, therefore, may be biased. Consequently, we use a hierarchical model that separates and uncovers both individual and aggregated effects via these random intercepts.

We are aware that these kind of models are referred to differently across various disciplines (e.g., mixed models, random effects models, hierarchical models, multilevel models...), which can lead to some confusion. Therefore, in our methods section we have chosen to use standard terms in hierarchical modelling theory. More specifically, we follow the well-established guide by Gelman and Hill [14]. Nevertheless, acknowledging that "multilevel modelling" is also a very common term (Gelman and Hill also mention this term in their book), we have added this clarification to our manuscript as well.

> If the function in Eq 1 is a logit model, then why the distribution of error is normal? That is not consistent with a logit model specification.

Please note that we do *not* use a logit model, and we do not make such a claim. Eq. 1 shows a logit function only because this is the link function required for considering this kind of response and error. As specified in the methods section, the model has a "fractional response" (the response is a proportion between 0 and 1, and the total counts are introduced as weights). In particular, it uses a quasi-binomial family (to account for overdispersion), which means that the error distribution of the weighted response (the proportion of males multiplied by the total number of cases, i.e., the number of males) is considered to be binomial with some overdispersion. This also means that, under these assumptions, the errors of the transformed response (logit of the proportion of males, as Eq. 1 shows) do follow a normal distribution. Therefore, our specification is consistent with fractional models.

For the sake of clarity, the revised version of the manuscript explains that we consider the quasi-binomial family as a description of the error distribution. Thank you for the observation.

> Why is p the proportion of males? The data is coming from individual articles. So what one would expect is for example p=1 if the paper has a male author (or a male first author, depending on what analysis you’re running). This is not aggregate model, so what is the reason for using the term proportion?

As note by Reviewer 2, "we estimate the monthly proportion of males" by category and subcategory for (1) "all authors", (2) "first authors", (3) "last authors", and (4) "single authors". We did this because there are short-term effects in which we are not interested and that would affect the estimation: e.g., there are fewer submissions during weekends and other holidays, and some distinctive patterns may arise even for weekday submissions. Therefore, monthly aggregates effectively mitigate these short-term artifacts.

> The most confusing part of this equation is the alpha coefficients, which have not even been elaborated on. Why such a confusing specification? What is the reason for the additional error terms for alphas? Those are going to confound with the main error term and the reviewer cannot even see the justification or interpretation of those.

Here again, we realize that the confusion may come from the fact that the model used has different names across different fields. As discussed in previous comments, Eq. 1 defines the nested structure of the random intercepts with the $\\alpha_{0, k[j[i]]}$ coefficient, where individual observations ($i$) are nested into subcategories ($j$), and those are nested into categories ($k$). Each level in this hierarchical structure has its own error term that must be specified and estimated separately. 

Aiming to avoid misunderstandings, we have added the "multilevel" term to the manuscript. Also, we would like to reiterate that we follow the well-established guide by Gelman and Hill in hierarchical/multilevel modelling [14], both in terms of the language and notation used. 

> Also this analysis (the logit analysis) is less problematic when we consider first authorships because a paper either has a male first author or a female first author (which is binary and consistent with a logit model specification). Therefore, if the coefficient for "year" for example is negative then one can conclude that the proportion of male authors have been decreasing over time (Although I am still not convinced why this has to be inferred from such an indirect complicated way as opposed to just reporting the proportion from the sample). But when it comes to "all author" analysis, then I am not sure how the models treats this. Is this where the issue of "proportion" comes to play? You count the proportion of male authors on the paper? Are you only using the papers for which the gender of all authors could be determined with 0.95 confidence?

Thank you for the remark. This observation relates to the points discussed in our previous comments, which we hope have clarified the main issues. Specifically, (1) our model is not logit but fractional, where the response is a proportion and not 0-1; and (2) the response is a proportion "for all cases", so we calculate (a) the monthly proportion of all authors that are males, (b) the monthly proportion of first authors that are males, (c) the monthly proportion of last authors that are males, and (d) the monthly proportion of single authors that are males. We hope this helps clarify the kind of model and response we use.

Regarding the "all authors" analysis, we refer to the section "Inferring gender from authors' given names", where we specify: "Following Holman et al., [2], we only consider the drafts with a probability higher or equal to $0.95$. Additionally, we excluded names that were not found more than $10$ times in the "genderize" database." Further on, in the "Measuring the effect of lockdown" section, we specify: "In the case of "all authors", preprints with missing gender rates greater than 25% are not considered, and subcategories with fewer than 30 authors per month are dropped too."

> Again, considering that the issue of lockdown was not a thing until 2020, I do not see the point of doing such joint/multivariate analysis, whereas, the proportion of males/females that published in 2020 could have just been reported independently. I apologise if this comment might sound too direct, but could it be that authors had to force themselves to use a more sophisticated statistical approach because simple reporting of proportions/stats could have come across rudimentary? Again, I apologise and I am sure that authors understand that the nature of a review sometimes entails direct questions of this nature. There is just limited interaction between these variables especially for 2017-2019 and that makes the use of such joint model very questionable to the reviewer. So could you justify the use of this method (and not choosing to just simply report the split of authorship in 2020 pre-prints)? Does it have to be inferred indirectly from a multivariate probabilistic model?

Comments are always welcome. We understand the nature of the review and appreciate any suggestions for improvement. 

In our view, the fact that lockdown was not an issue until 2020 is the main reason why we do need multivariate probabilistic modelling. Previous work already reported descriptive values about the proportion of males/females in 2020, but this provides a very limited picture for several reasons.

First, Figure 2 shows an overall trend whereby we expect to see more women every year participating in preprint submissions. Yet, a reversal of this pattern is observed during confinement. If we only reported the proportion of male/female authors during the lockdown months, how much of the gender gap could be attributed to the lockdown effect? How much of it could be simply explained by the temporal trend? (i.e., what we would have expected without a pandemic). By expanding the observation window, we are able to discern how much of the observed change in 2020 corresponds to the effect of lockdown and social distancing and how much is due to gender differences in the circulation of scientific knowledge over time [15-17]. 

Second, the gender trend in preprint submissions is not homogeneous: different fields/sub-fields have historically attracted more/less women, and thus they present distinctive patterns. 

Finally, preprint submissions have experienced an unprecedented growth, in part due to the emergence of COVID-related research (see Figure 1). Does COVID-related research display different gender patterns than "general" research?

Our model allows us to answer all these questions and provides a much better understanding of the impact of COVID-19 on the gender gap compared to previous work reporting descriptive data. A detailed discussion of the empirical and substantive contribution of our research can be found both in the introduction and conclusion of the manuscript.

In the absence of captions for Supplementary figures, the reviewer unfortunately has no idea how to interpret them.

Please note that PLOS ONE requires supplementary tables and figures to be uploaded separately, while their captions can be found in the manuscript in a section called "Supporting information" that is located between our conclusions and references. We understand the confusion this may have caused, but this is how we are required to format the materials according to the journal guidelines.

# References

------------

[1]Santamaría L, Mihaljevi ´c H. Comparison and benchmark of name-to-gender inference services. PeerJComputer Science. 2018;4:e156.

[2]Holman L, Stuart-Fox D, Hauser CE. The gender gap in science: How long until women are equallyrepresented? PLOS Biology. 2018 Apr;16(4):e2004956. Available from: https://doi.org/10.1371/journal.pbio.2004956.

[3]Myers KR, Tham WY, Yin Y, Cohodes N, Thursby JG, Thursby MC, et al. Unequal effects of theCOVID-19 pandemic on scientists. Nature human behaviour. 2020;4(9):880–883.

[4]Andersen JP, Nielsen MW, Simone NL, Lewiss RE, Jagsi R. Meta-Research: COVID-19 medical papershave fewer women first authors than expected. elife. 2020;9:e58807.7

[5]Barber BM, Jiang W, Morse A, Puri M, Tookes H, Werner IM. What Explains Differences in FinanceResearch Productivity During the Pandemic? The Journal of Finance. 2021.

[6]Staniscuaski F, Kmetzsch L, Soletti RC, Reichert F, Zandonà E, Ludwig ZM, et al. Gender, race andparenthood impact academic productivity during the COVID-19 pandemic: from survey to action.Frontiers in psychology. 2021;12.

[7]Gayet-Ageron A, Messaoud KB, Richards M, Schroter S. Female authorship of covid-19 research inmanuscripts submitted to 11 biomedical journals: cross sectional study. BMJ. 2021;375.

[8]Ribarovska AK, Hutchinson MR, Pittman QJ, Pariante C, Spencer SJ. Gender inequality in publishingduring the COVID-19 pandemic. Brain, behavior, and immunity. 2021;91:1.

[9]Harrop C, Bal V, Carpenter K, Halladay A. A lost generation? The impact of the COVID-19 pandemicon early career ASD researchers. Autism Research. 2021.

[10]Oleschuk M. Gender equity considerations for tenure and promotion during COVID-19. Canadianreview of sociology. 2020.

[11]Viglione G. Are women publishing less during the pandemic? Here’s what the data say. Nature.2020;581:365–366. Available from: https://www.nature.com/articles/d41586-020-01294-9.

[12]Matthews D. Pandemic lockdown holding back female academics, data show. Times Higher Education.2020. Available from: https://www.timeshighereducation.com/news/pandemic-lockdown-holding-back-female-academics-data-show.

[13]Klebel T, Reichmann S, Polka J, McDowell G, Penfold N, Hindle S, et al. Peer review and preprintpolicies are unclear at most major journals. PLOS ONE. 2020 10;15(10):1–19. Available from: https://doi.org/10.1371/journal.pone.0239518.

[14]Gelman A, Hill J. Data Analysis Using Regression and Multilevel/Hierarchical Models. AnalyticalMethods for Social Research. Cambridge University Press; 2006.

[15]Zhu Y. Who support open access publishing? Gender, discipline, seniority and other factors associatedwith academics’ OA practice. Scientometrics. 2017;111(2):557–579. Available from: https://doi.org/10.1007/s11192-017-2316-z.

[16]J T, A K, J M, C R, E MH, C P, et al. Gender differences and bias in open source: pull request acceptanceof women versus men. PeerJ Computer Science 3:e111. 2017.

[17]Ruggieri R, Pecoraro F, Luzi D. An intersectional approach to analyse gender productivity and openaccess: a bibliometric analysis of the Italian National Research Council. Scientometrics. 2021;126(2):1647–1673.

---

## [Decision Letter · Decision Letter 1]

2 Feb 2022

PONE-D-21-25133R1Mind the gender gap: COVID-19 lockdown effects on gender differences in preprint submissionsPLOS ONE

Dear Dr. Ucar,

Thank you for submitting your manuscript to PLOS ONE. After careful consideration, we feel that it has merit but does not fully meet PLOS ONE’s publication criteria as it currently stands. Therefore, we invite you to submit a revised version of the manuscript that addresses the points raised during the review process.

We look forward to receiving your revised manuscript.

Kind regards,

Alireza Abbasi

Academic Editor

PLOS ONE

Journal Requirements:

Reviewers' comments:

Reviewer's Responses to Questions

**Comments to the Author**

1. If the authors have adequately addressed your comments raised in a previous round of review and you feel that this manuscript is now acceptable for publication, you may indicate that here to bypass the “Comments to the Author” section, enter your conflict of interest statement in the “Confidential to Editor” section, and submit your "Accept" recommendation.

Reviewer #1: All comments have been addressed

2. Is the manuscript technically sound, and do the data support the conclusions?

Reviewer #1: Yes

3. Has the statistical analysis been performed appropriately and rigorously? 

Reviewer #1: I Don't Know

4. Have the authors made all data underlying the findings in their manuscript fully available?

Reviewer #1: Yes

5. Is the manuscript presented in an intelligible fashion and written in standard English?

Reviewer #1: Yes

6. Review Comments to the Author

Reviewer #1: The authors have provided detailed responses to reviewers' comments and have made appropriate amendments. The only thing that was not clear to me in the revised format was whether the change in the number of preprints (changes in data presented in table 1) resulted in any change in the results of the regression analysis. I see they are the same as before while the number of preprints has changed. I just want the authors to ensure that there hasn't been an oversight in relation to this and the results are not erroneous.

7. PLOS authors have the option to publish the peer review history of their article (what does this mean?). If published, this will include your full peer review and any attached files.

Reviewer #1: **Yes: **Hamid R. Jamali

---

## [Author Response · Author response to Decision Letter 1]

5 Feb 2022

We would like to thank the reviewers again for their time and valuable comments, which have been a great help in improving the paper. According to the feedback from revision 1, we are required to address one last comment:

> Reviewer 1: The authors have provided detailed responses to reviewers' comments and have made appropriate amendments. The only thing that was not clear to me in the revised format was whether the change in the number of preprints (changes in data presented in table 1) resulted in any change in the results of the regression analysis. I see they are the same as before while the number of preprints has changed. I just want the authors to ensure that there hasn't been an oversight in relation to this and the results are not erroneous.

We confirm that the results of the regression analysis were correct and didn't change. It was just a mistake in the counts provided in our first version of Table 1, but the models were fed with the correct data, and the results have been checked several times. Accordingly, we proceeded to submit this new revision with no changes.

---

## [Decision Letter · Decision Letter 2]

8 Feb 2022

Mind the gender gap: COVID-19 lockdown effects on gender differences in preprint submissions

PONE-D-21-25133R2

Dear Dr. Ucar,

We’re pleased to inform you that your manuscript has been judged scientifically suitable for publication and will be formally accepted for publication once it meets all outstanding technical requirements.

Kind regards,

Ali B. Mahmoud, Ph.D.

Academic Editor

PLOS ONE

Additional Editor Comments (optional):

Reviewers' comments:

Reviewer's Responses to Questions

**Comments to the Author**

1. If the authors have adequately addressed your comments raised in a previous round of review and you feel that this manuscript is now acceptable for publication, you may indicate that here to bypass the “Comments to the Author” section, enter your conflict of interest statement in the “Confidential to Editor” section, and submit your "Accept" recommendation.

Reviewer #1: All comments have been addressed

2. Is the manuscript technically sound, and do the data support the conclusions?

Reviewer #1: Yes

3. Has the statistical analysis been performed appropriately and rigorously? 

Reviewer #1: Yes

4. Have the authors made all data underlying the findings in their manuscript fully available?

Reviewer #1: Yes

5. Is the manuscript presented in an intelligible fashion and written in standard English?

Reviewer #1: Yes

6. Review Comments to the Author

Reviewer #1: Happy with the response of the authors about the accuracy of their analysis. The manuscript is ok for publishing.

7. PLOS authors have the option to publish the peer review history of their article (what does this mean?). If published, this will include your full peer review and any attached files.

Reviewer #1: **Yes: **Hamid R. Jamali

---

## [Editor Report · Acceptance letter]

14 Feb 2022

PONE-D-21-25133R2 

Mind the gender gap: COVID-19 lockdown effects on gender differences in preprint submissions 

Dear Dr. Ucar:

I'm pleased to inform you that your manuscript has been deemed suitable for publication in PLOS ONE. Congratulations! Your manuscript is now with our production department. 

Kind regards, 

on behalf of

Dr. Ali B. Mahmoud 

Academic Editor

PLOS ONE